# DNA: Denoised Neighborhood Aggregation for Fine-grained Category Discovery

**Wenbin An**[1], **Feng Tian**[2*],**Wenkai Shi**[1], **Yan Chen**[2], **Qinghua Zheng**[2]
**QianYing Wang**[3], **Ping Chen**[4]

[1] School of Automation Science and Engineering, Xi'an Jiaotong University
[2] School of Computer Science and Technology, MOEKLNNS Lab, Xi'an Jiaotong University
[3] Lenovo Research [4] Department of Engineering, University of Massachusetts Boston
wenbinan@stu.xjtu.edu.cn,{fengtian,chenyan}@mail.xjtu.edu.cn
shiyibai778@gmail.com,wangqya@lenovo.com, ping.chen@umb.edu

## Abstract

Discovering fine-grained categories from coarsely labeled data is a practical and challenging task, which can bridge the gap between the demand for fine-grained analysis and the high annotation cost. Previous works mainly focus on instance-level discrimination to learn low-level features, but ignore semantic similarities between data, which may prevent these models learning compact cluster representations. In this paper, we propose *Denoised Neighborhood Aggregation* (DNA), a self-supervised framework that encodes semantic structures of data into the embedding space. Specifically, we retrieve $k$-nearest neighbors of a query as its positive keys to capture semantic similarities between data and then aggregate information from the neighbors to learn compact cluster representations, which can make fine-grained categories more separatable. However, the retrieved neighbors can be noisy and contain many false-positive keys, which can degrade the quality of learned embeddings. To cope with this challenge, we propose three principles to filter out these false neighbors for better representation learning. Furthermore, we theoretically justify that the learning objective of our framework is equivalent to a clustering loss, which can capture semantic similarities between data to form compact fine-grained clusters. Extensive experiments on three benchmark datasets show that our method can retrieve more accurate neighbors (21.31% accuracy improvement) and outperform state-of-the-art models by a large margin (average 9.96% improvement on three metrics). Our code and data are available at https://github.com/Lackel/DNA.

## 1 Introduction

Many AI fields have progressed into fine-grained analysis, e.g., Computer Vision (Wei et al., 2021;

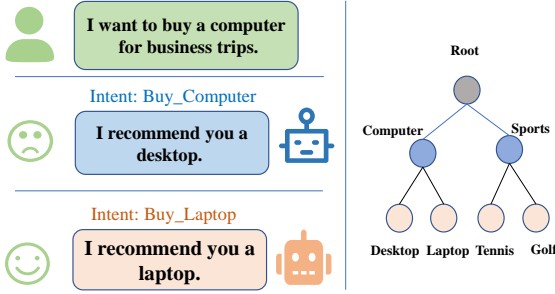

Figure 1: **Left**: An example of coarse- and fine-grained intent detection for recommendation. **Right**: Label hierarchy with coarse- and fine-grained categories.

Nauta et al., 2021) and Natural Language Processing (Suresh and Ong, 2021; Vaid et al., 2022; Munikar et al., 2019; Almeida et al., 2021), since it can provide much more information than coarse-grained analysis. For example, detecting more fine-grained user intents can help to provide more accurate recommendation and better services for customers (Figure 1 Left). However, labelling fine-grained categories can be time-consuming and labour-intensive since it requires more expert knowledge. To get out of this dilemma, a novel task called Fine-grained Category Discovery under Coarse-grained supervision (FCDC) was recently proposed by An et al. (2022a). Taking Figure 1 Right as an example, FCDC aims at discovering fine-grained categories (e.g., Desktop and Tennis) using only coarse-grained (e.g., Computer and Sports) labeled data which are easier and cheaper to annotate.

To solve the FCDC task, previous methods mainly focus on instance-level discrimination to learn low-level features through contrastive learning (An et al., 2022a; Bukchin et al., 2021). Despite the improved performance, these instance-based methods fail to encode cluster-level semantic structures of data. This is because these methods simply treat each instance as a single class and push away other instances, regardless of their se-

---

*Corresponding Author.

mantic similarities (Li et al., 2020), which can hinder the formation of compact fine-grained clusters. Here we define 'compact' as samples with the same fine-grained categories are compactly clustered into the center of category and away from other samples from different fine-grained categories, which means smaller intra-class distance and larger inter-class distance. Since samples located around decision boundaries are easily misclassified into wrong categories, distributing samples near the category center compactly can avoid overlapping decision boundaries and make these categories more distinguishable. So learning compact cluster representations is important for the FCDC task to learn more separable fine-grained categories.

To encode semantic structures of data to learn more compact cluster representations, we propose a novel model named *Denoised Neighborhood Aggregation* (DNA). DNA can capture semantic similarities between data by retrieving $k$-nearest neighbors of a query and aggregating information from them. However, the retrieved neighbors can be noisy and contain many false-positive keys (i.e., keys with different fine-grained categories from the query), which can reduce the quality of representation learning. This situation is more severe in the FCDC setting since pretraining on coarse-grained labels can easily include wrong neighbors for those samples with the same coarse-grained labels but different fine-grained ones. To solve this problem, we propose three principles (named Label Constraint, Reciprocal Constraint, and Rank Statistic Constraint) to filter out these false neighbors. These constraints consider bidirectional semantic structures and statistical features of data to help to retrieve more accurate neighbors. Furthermore, we interpret our framework from a generalized Expectation-Maximization (EM) perspective. At the E-step, we retrieve reliable neighbors from a dynamic queue under the proposed constraints, then at the M-step, we perform neighborhood aggregation to encode semantic structures of data to learn more compact representations. Last but not least, we theoretically prove that the learning objective of our model is equivalent to a clustering loss, which can help to learn compact cluster representations to facilitate fine-grained category discovery.

Our main contributions can be summarized as follows:

- **Perspective**: we propose to model semantic structures of data to learn more compact cluster representations, which are essential for the FCDC task.

- **Framework**: we propose *Denoised Neighborhood Aggregation*, a self-supervised framework that captures semantic similarities between data and aggregates information from neighbors. We further propose three principles to filter out false neighbors for better representation learning.

- **Theory**: we interpret our framework from a generalized EM perspective and theoretically prove that the learning objective of our framework is equivalent to a clustering loss. So our model can alternately retrieve more accurate neighbors and learn more compact cluster representations.

- **Experiments**: Extensive experiments on three benchmark datasets show that our model establishes state-of-the-art performance on the FCDC task (average 9.96% improvement) and retrieves more accurate neighbors (21.31% accuracy improvement), which validates our theoretical analysis.

## 2 Related Work

### 2.1 Novel Category Discovery

Novel Category Discovery aims at discovering novel categories from unlabeled data to expand existing class taxonomy (Scheirer et al., 2014; Zhang et al., 2021a; Vaze et al., 2022; Yu et al., 2022; Badirli et al., 2023; An et al., 2023). To discover novel categories without any annotation, previous models usually adopted self-supervised methods. For example, Han et al. (2020) utilized ranking statistics as pseudo-labels to train their model with binary cross-entropy loss. An et al. (2022b) proposed to decouple known and novel categories from unlabeled data and performed representation learning with prototypical network. However, these methods only focus on the scenario where known and novel categories are of the same granularity. To discover fine-grained categories, a novel task called Fine-grained Category Discovery under Coarse-grained supervision (FCDC) was proposed by An et al. (2022a). They also proposed a weighted self-contrastive strategy to acquire fine-grained knowledge. And Mekala et al. (2021) proposed to perform fine-grained text classification with the help

of fine-grained label names and coarse-grained labeled data. In Computer Vision, Bukchin et al. (2021) proposed angular contrastive learning to perform few-shot fine-grained image classification with only coarse-grained supervision. However, these methods only focus on instance-level discrimination, which may prevent them from learning compact cluster representations for fine-grained category discovery.

## 2.2 Contrastive Learning

Contrastive Learning (CL) performs representation learning by pulling similar samples closer and pushing dissimilar samples far away (Chen et al., 2020). And how to build high-quality positive keys for the given queries is a challenging task for CL. Most previous methods took two different transformations of the same input as query and positive key, respectively (Dosovitskiy et al., 2014; Chen et al., 2020; He et al., 2020). Li et al. (2020) proposed to utilize prototypes learned by clustering as their positive keys. Furthermore, An et al. (2022a) proposed to use shallow features extracted by BERT as positive keys. Recently, Neighbourhood Contrastive Learning (NCL) was proposed by treating the nearest neighbors of queries as positive keys (Dwibedi et al., 2021a), which can avoid complex data augmentations. Zhong et al. (2021) further utilized $k$-nearest neighbors to mine hard negative keys for CL. And Zhang et al. (2022a) randomly selected one positive key from $k$-nearest neighbors for representation learning. Even though NCL has achieved better results on many tasks, previous methods ignored the fact that the retrieved neighbors can be noisy (i.e., neighbors and the query come from different categories) due to lack of supervision, and these false-positive keys can be harmful for representation learning since they provide wrong supervision signals.

## 3 Method

### 3.1 Problem Formulation

Given a set of coarse-grained categories $\mathcal{Y}_{coarse} = \{\mathcal{C}_1, \mathcal{C}_2, ..., \mathcal{C}_M\}$ and a coarsely labeled training set $\mathcal{D}_{train} = \{(x_i, c_i) \mid c_i \in \mathcal{Y}_{coarse}\}_{i=1}^{N}$, the FCDC task aims at learning a feature encoder $F_\theta$ that maps samples into a compact $D$-dimension embedding space to further separate them into different fine-grained categories $\mathcal{Y}_{fine} = \{\mathcal{F}_1, \mathcal{F}_2, ..., \mathcal{F}_K\}$, even though without any prior fine-grained knowledge, where $\mathcal{Y}_{fine}$ are sub-classes of $\mathcal{Y}_{coarse}$.

Model performance will be measured on another testing set $\mathcal{D}_{test} = \{(x_i, y_i) \mid y_i \in \mathcal{Y}_{fine}\}_{i=1}^{L}$ through clustering (e.g., K-Means) based on the embeddings extracted by $F_\theta$. It should be noted that we only use the number of fine-grained categories $K$ when testing so that we can make a fair comparison with different methods, following the settings in previous work (An et al., 2022a).

### 3.2 Proposed Approach

To achieve the learning objective of FCDC, we propose *Denoised Neighborhood Aggregation* (DNA), an iterative framework to bootstrap model performance on retrieving reliable neighbors and learning compact embeddings. As shown in Fig. 2, our model mainly contains three steps. Firstly, we maintain a dynamic queue to retrieve neighbors for queries based on their semantic similarities (Sec. 3.2.1). Secondly, we propose three principles to filter out false-positive neighbors for better representation learning (Sec. 3.2.2). Thirdly, we perform neighborhood aggregation to learn compact embeddings for fine-grained clusters (Sec. 3.2.3). Last but not least, we interpret our framework from the generalized EM algorithm perspective and theoretically prove that our learning objective is equivalent to a clustering loss, which can help to learn more compact fine-grained cluster embeddings (Sec. 3.2.4).

### 3.2.1 Neighborhood Retrieval

We maintain a dynamic queue $\mathcal{M}$ to store sample features for subsequent training. The features in the queue are extracted by a momentum encoder $F_{\theta^m}$ and are progressively updated at each iteration. To keep consistency of features used for neighborhood retrieval and representation learning, we update $F_{\theta^m}$ in a moving-average manner (He et al., 2020):

$$\theta_{t+1}^m = \alpha\theta_t^m + (1-\alpha)\theta_{t+1} \tag{1}$$

where $\alpha \in [0, 1)$ is a momentum coefficient, $\theta^m$ and $\theta$ are parameters of $F_{\theta^m}$ and $F_\theta$, respectively. $F_\theta$ is a query encoder for representation learning and is updated by back-propagation.

In order to learn compact representations, we retrieve neighbors of each query from the queue $\mathcal{M}$. Specifically, we first pretrain $F_\theta$ and $F_{\theta^m}$ with cross-entropy loss on coarse-grained labels to initialize models. Then given a query embedding $q_i = F_\theta(x_i)$, we search its $k$-nearest neighbors $\mathcal{N}_i$ from the queue $\mathcal{M}$ by measuring their semantic

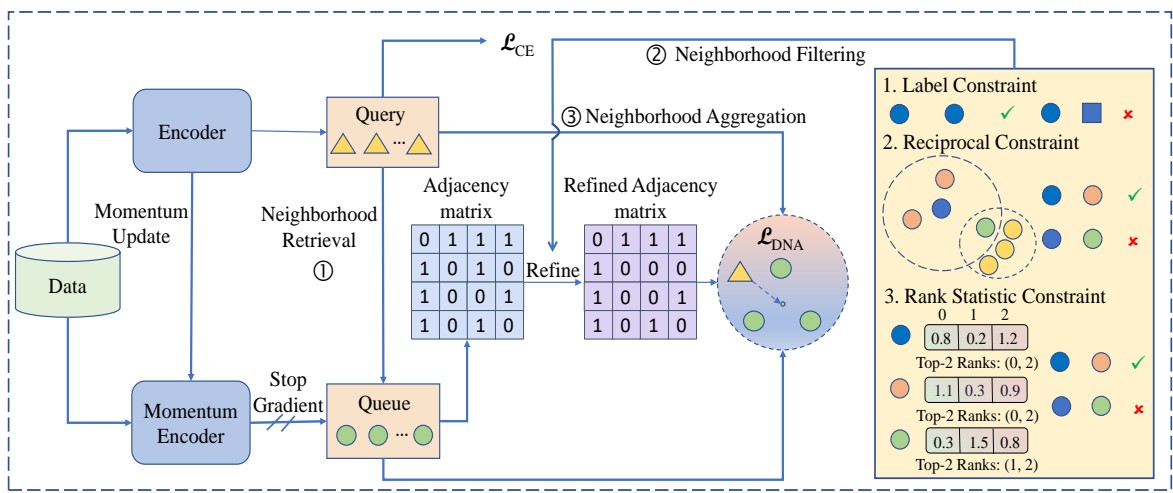

Figure 2: The overall architecture of our *DNA* framework.

similarities:

$$\mathcal{N}_i = \{h_j \mid h_j \in \underset{h_l \in \mathcal{M}}{argtop_k}(sim(q_i, h_l))\} \quad (2)$$

where $sim()$ is a similarity function and here we use cosine similarity $sim(q_i, h_j) = \frac{q_i^T h_j}{\|q_i\| \cdot \|h_j\|}$.

### 3.2.2 Neighborhood Refining

After neighborhood retrieval, previous methods (Dwibedi et al., 2021a; Zhang et al., 2022a) simply used these neighbors as positive keys for contrastive learning. However, they ignored the fact that the retrieved neighbors contain many false-positive keys (i.e., keys with different fine-grained categories from the query), which can significantly degrade their model performance. This is because we lack of fine-grained supervision and samples with different fine-grained categories can be clustered together after pretraining. To mitigate this problem, we propose three principles to filter out these false-positive neighbors, which consider bidirectional semantic structures and statistical features of samples (illustrated in Fig. 2).

**Label Constraint** aims at filtering neighbors with different coarse-grained labels from the query, since samples with the same fine-grained labels must also have the same coarse-grained ones. The refined neighbor set for query $q_i$ is:

$$\mathcal{A}_i = \{h_j \mid (h_j \in \mathcal{N}_i) \land (c_i = c_j)\} \quad (3)$$

where $c_i$ and $c_j$ are coarse-grained labels for the query $q_i$ and its neighbor $h_j$.

**Reciprocal Constraint** requires that the neighborhood relationships should be bidirectional (Qin et al., 2011) (i.e., the query should also be its neighbors' neighbor). This constraint is intuitive since samples in a compact cluster should be neighbors to each other. Traditional $k$-NN simply retrieved $k$ neighbors for each sample, which can introduce many false neighbors, especially for data with long-tailed distribution. As the example of reciprocal constraint in Fig. 2, the blue sample should only have two neighbors, but 3-NN introduces a false neighbor (the green sample) for it. The reciprocal constraint can filter out this false neighbor since the blue sample is not a neighbor of the green one. So the reciprocal constraint can capture bidirectional semantic structures between samples and provide different number of neighbors for different samples. And the refined neighbor set for query $q_i$ is:

$$\mathcal{R}_i = \{h_j \mid (h_j \in \mathcal{A}_i) \land (q_i \in \mathcal{A}_j)\} \quad (4)$$

**Rank Statistic Constraint** requires the rank statistics between neighbors to be the same. Specifically, we rank the values of feature embeddings by magnitude and extract the index of top-$m$ values to form a rank set. Then we filter out neighbors who have different rank set from the query. The rank statistic constraint is effective for two reasons. Firstly, rank statistic is more robust than cosine similarity, especially for high-dimensional data (Friedman, 1994; Han et al., 2020). Secondly, samples with the same coarse-grained labels but different fine-grained ones can have high cosine similarities after pretraining, which can lead to false-positive neighbors. However, we think the pretrained embeddings also contain information about fine-grained categories which are interrupted by other noisy information. So if we only focus on the main components

of these embeddings, we can filter out the noisy information and discover the hidden fine-grained information. The refined neighbor set for query $q_i$ is:

$$\mathcal{S}_i = \{h_j \mid (h_j \in \mathcal{R}_i) \wedge (top_m(h_j) = top_m(q_i))\} \tag{5}$$

where $top_m(q_i)$ maps the $d$-dimensional embedding $q_i$ into a $m$-dimensional set ($m = 2$ in Fig. 2) which contains index of top-$m$ values of $q_i$.

### 3.2.3 Denoised Neighborhood Aggregation

After mining reliable neighbors, we perform *Denoised Neighborhood Aggregation* (DNA) to pull queries and their neighbors closer by extending the traditional contrastive loss (Oord et al., 2018) to the form with multiple positive keys:

$$
\begin{aligned}
\mathcal{L}_{DNA} &= -\frac{1}{|\mathcal{D}|} \sum_{q_i \in \mathcal{D}} \frac{1}{|\mathcal{S}_i|} \sum_{h_j \in \mathcal{S}_i} log \frac{exp(q_i^T h_j / \tau)}{\sum_{h_k \in \mathcal{M}} exp(q_i^T h_k / \tau)} \\
&= \underbrace{-\frac{1}{|\mathcal{D}|} \sum_{q_i \in \mathcal{D}} \frac{1}{|\mathcal{S}_i|} \sum_{h_j \in \mathcal{S}_i} (q_i^T h_j / \tau)}_{(Alignment)} \\
&\quad + \underbrace{\frac{1}{|\mathcal{D}|} \sum_{q_i \in \mathcal{D}} log \sum_{h_k \in \mathcal{M}} exp(q_i^T h_k / \tau)}_{(Uniformity)}
\end{aligned}
\tag{6}
$$

where $q_i$ is the $L_2$ normalized query embedding and $h_j$ is the $L_2$ normalized key embedding from the queue $\mathcal{M}$. Then we train the model with the loss $\mathcal{L}_{DNA}$ and the cross-entropy loss $\mathcal{L}_{CE}$ with the coarse-grained labels to learn compact cluster embeddings for FCDC.

### 3.2.4 Theoretical Analysis

**Analysis for $\mathcal{L}_{DNA}$.** The loss $\mathcal{L}_{DNA}$ can be divided into two parts (Wang and Isola, 2020): *Alignment* to pull queries and their neighbors closer, and *Uniformity* to make samples uniformly distributed in hyper-sphere. Then we will prove that the *Alignment* term in $\mathcal{L}_{DNA}$ is equivalent to a clustering loss that makes the query converge to the center of its neighbors, which can help to learn more compact cluster representations for fine-grained cate-

gory discovery.

$$
\begin{aligned}
(Align.) &= -\frac{1}{\tau|\mathcal{D}|} \sum_{q_i \in \mathcal{D}} \left\{ q_i^T \left( \frac{1}{|\mathcal{S}_i|} \sum_{h_j \in \mathcal{S}_i} h_j \right) \right\} \\
&= -\frac{1}{\tau|\mathcal{D}|} \sum_{q_i \in \mathcal{D}} q_i^T \mu_i \\
&= \frac{1}{2\tau|\mathcal{D}|} \sum_{q_i \in \mathcal{D}} \left\{ (q_i - \mu_i)^2 - \|q_i\|^2 - \|\mu_i\|^2 \right\} \\
&= \frac{1}{2\tau|\mathcal{D}|} \sum_{q_i \in \mathcal{D}} (q_i - \mu_i)^2 + c \\
&\overset{c}{=} \sum_{q_i \in \mathcal{D}} (q_i - \mu_i)^2
\end{aligned}
\tag{7}
$$

where the symbol $\overset{c}{=}$ indicates equal up to a multiplicative and/or an additive constant. $\mu_i = \frac{1}{|\mathcal{S}_i|} \sum_{h_j \in \mathcal{S}_i} h_j$ is the center of query's neighbors. $\|q_i\|^2 = 1$ because of normalization, and $\|\mu_i\|^2$ is a constant since the neighbor embedding $h_j$ is from the queue without gradient.

**Interpreting *DNA* from the generalized EM perspective.** If we consider the centers of the neighbors as hidden variables, we can interpret the *DNA* framework from the generalized EM perspective.

At the E-step, we fix model parameters $\theta$ to find the hidden variables by retrieving and refining the neighbor set $\mathcal{S}_i$:

$$\{\mu_i | \theta, q_i, \mathcal{M}\}_{i=1}^N = \frac{1}{|\mathcal{S}_i|} \sum_{h_j \in \mathcal{S}_i} h_j \tag{8}$$

At the M-step, we fix the hidden variables $\{\mu_i\}_{i=1}^N$ to optimize model parameters $\theta$:

$$\underset{\theta}{argmin} \sum_{q_i \in \mathcal{D}} (q_i - \mu_i)^2 \tag{9}$$

Since more accurate neighbors can boost representation learning and better representation learning can help to retrieve more accurate neighbors, our framework can iteratively bootstrap model performance on representation learning and neighborhood retrieval. In addition to the intuitive explanation, we also verify the intuition through experiments (Sec. 5.4).

## 4 Experiments

### 4.1 Experimental Settings

#### 4.1.1 Datasets

We conduct experiments on three benchmark datasets. **CLINC** (Larson et al., 2019) is an intent

| Dataset | $|\mathcal{C}|$ | $|\mathcal{F}|$ | # Train | # Test |
|---------|------|------|---------|--------|
| CLINC | 10 | 150 | 18,000 | 1,000 |
| WOS | 7 | 33 | 8,362 | 2,420 |
| HWU64 | 18 | 64 | 8,954 | 1,031 |

Table 1: Statistics of benchmark datasets. #: number of samples. $|\mathcal{C}|$: number of coarse-grained categories. $|\mathcal{F}|$: number of fine-grained categories.

detection dataset from multiple domains. **WOS** (Kowsari et al., 2017) is a paper classification dataset. **HWU64** (Liu et al., 2021) is an assistant query classification dataset. Statistics of the datasets are shown in Table 1.

### 4.1.2 SOTA Methods for Comparison

We compare our model with following methods. **Baselines**: BERT (Devlin et al., 2018) without fine-tuning and BERT under coarse-grained supervision. **Self-training Methods**: DeepCluster (Caron et al., 2018) and DeepAligned (Zhang et al., 2021b). **Contrastive based Methods**: SimCSE (Gao et al., 2021), Ancor (Bukchin et al., 2021), Delete (Wu et al., 2020), Nearest-Neighbor Contrastive Learning (NNCL) (Dwibedi et al., 2021b), Contrastive Learning with Nearest Neighbors (CLNN) (Zhang et al., 2022b) and Weighted Self-Contrastive Learning (WSCL) (An et al., 2022a). We also investigate some variants by adding cross-entropy loss (**+CE**).

### 4.2 Evaluation Metrics

To evaluate the quality of the discovered fine-grained clusters, we use two broadly used evaluation metrics: Adjusted Rand Index (ARI) (Hubert and Arabie, 1985) and Normalized Mutual Information (NMI) (Lancichinetti et al., 2009):

$$ARI = \frac{RI - E(RI)}{max(RI) - E(RI)} \quad (10)$$

$$NMI = \frac{2 * I(\hat{y}; y)}{H(\hat{y}) + H(y)} \quad (11)$$

where $RI$ is the rand index and $E(RI)$ is the expectation of $RI$. $\hat{y}$ is the prediction from clustering and $y$ is the ground truth. $I(\hat{y}; y)$ is the mutual information between $\hat{y}$ and $y$, $H(\hat{y})$ and $H(y)$ represent the entropy of $\hat{y}$ and $y$, respectively.

To evaluate the classification performance of models, we use the metric clustering accuracy (ACC):

$$ACC = \frac{\sum_{i=1}^{N} \mathbb{I}\{\mathcal{P}(\hat{y}_i) = y_i\}}{N} \quad (12)$$

where $\hat{y}_i$ is the prediction from clustering and $y_i$ is the ground-truth label, $N$ is the number of samples, $\mathcal{P}(\cdot)$ is the permutation map function from Hungarian algorithm (Kuhn, 1955).

### 4.2.1 Implementation Details

We use the pre-trained BERT-base model (Devlin et al., 2018) as our backbone with the learning rate $5e^{-5}$. We use the AdamW optimizer with 0.01 weight decay and 1.0 gradient clipping. For hyper-parameters, the batch size for pretraining, training and testing is set to 64. Epochs for pretraining and training are set to 100 and 20, respectively. The temperature $\tau$ is set to 0.07. The number of neighbors $k$ is set to {120, 120, 250} for the dataset CLINC, HWU64 and WOS, respectively. The dimension for Rank Statistic Constraint is set to 5. The momentum factor $\alpha$ is set to 0.99. For compared methods, we use the same BERT model as ours to extract features and adopt hyper-parameters in their original papers for a fair comparison.

### 4.3 Result Analysis

Comparison results of different methods are shown in Table 2. From the results, we can get following observations. (1) Our model outperforms the compared methods across all evaluation metrics and datasets, which clearly shows the effectiveness of our model. we attribute the better performance of our model to the following reasons. Firstly, our model can model semantic structures of data to learn more compact cluster representations by aggregating information from neighbors. Secondly, our model can retrieve reliable neighbors to boost the quality of representation learning with three filtering principles. Thirdly, the previous two steps can boost performance of each other in an EM manner, which can progressively bootstrap the entire model performance. (2) Baselines and self-training methods perform badly on the FCDC task since they rely on abundant fine-grained labeled data to train their models, which are not available under the FCDC setting. (3) Contrastive-based methods perform better than baselines above since they can acquire fine-grained knowledge even without fine-grained supervision. However, these methods simply treat each instance as a single class but ignore semantic similarities between different instances, which may prevent them from learning compact representations for subsequent clustering.

| Methods | CLINC | | | WOS | | | HWU64 | | |
|---|---|---|---|---|---|---|---|---|---|
| | ACC | ARI | NMI | ACC | ARI | NMI | ACC | ARI | NMI |
| BERT | 34.37 | 17.61 | 64.75 | 31.97 | 18.36 | 45.15 | 33.52 | 17.04 | 56.90 |
| BERT + CE | 43.85 | 32.37 | 78.58 | 38.29 | 36.94 | 64.72 | 37.89 | 33.68 | 74.63 |
| DeepCluster | 26.40 | 12.51 | 61.26 | 29.17 | 18.05 | 43.34 | 29.74 | 13.98 | 53.27 |
| DeepAligned | 29.16 | 14.15 | 62.78 | 28.47 | 15.94 | 43.52 | 29.14 | 12.89 | 52.99 |
| DeepCluster + CE | 30.28 | 13.56 | 62.38 | 38.76 | 35.21 | 60.30 | 41.73 | 27.81 | 66.81 |
| DeepAligned + CE | 42.09 | 28.09 | 72.78 | 39.42 | 33.67 | 61.60 | 42.19 | 28.15 | 66.50 |
| NNCL | 17.42 | 13.93 | 67.56 | 29.64 | 28.51 | 61.37 | 32.98 | 30.02 | 73.24 |
| CLNN | 19.96 | 14.76 | 68.30 | 29.48 | 28.42 | 60.99 | 37.21 | 34.66 | 75.27 |
| SimCSE | 40.22 | 23.57 | 69.02 | 25.87 | 13.03 | 38.53 | 24.48 | 8.42 | 46.94 |
| Ancor + CE | 44.44 | 31.50 | 74.67 | 39.34 | 26.14 | 54.35 | 32.90 | 30.71 | 74.73 |
| Ancor | 45.60 | 33.11 | 75.23 | 41.20 | 37.00 | 65.42 | 37.34 | 34.75 | 74.99 |
| Delete | 47.11 | 31.28 | 73.39 | 24.50 | 11.68 | 35.47 | 21.30 | 6.52 | 44.13 |
| Delete + CE | 47.87 | 33.79 | 76.25 | 41.53 | 33.78 | 61.01 | 35.13 | 31.84 | 74.88 |
| SimCSE + CE | 52.53 | 37.03 | 77.39 | 41.28 | 34.47 | 61.62 | 34.04 | 31.81 | 74.86 |
| WSCL | 74.02 | 62.98 | 88.37 | 65.27 | 51.78 | 72.46 | 59.52 | 49.34 | 79.31 |
| **Ours** | **87.66** | **81.82** | **94.69** | **74.57** | **63.30** | **76.86** | **70.81** | **59.66** | **83.31** |
| Improvement | +13.64 | +18.84 | +6.32 | +9.30 | +11.52 | +4.40 | +11.29 | +10.32 | +4.00 |

Table 2: Model performance (%) on the FCDC task. Average results over 3 runs are reported. Some results are cited from An et al. (2022a).

## 5 Discussion

### 5.1 Ablation Study

We investigate the effectiveness of different components of our model in Table 3. From the table we can draw following conclusions. (1) Traditional Nearest-Neighbor Contrastive Learning (NNCL) performs badly on the FCDC task, which is because NNCL ignores semantic similarities between samples and fails to retrieve reliable neighbors. (2) Adding multiple neighbors as positive keys (Eq. 6) significantly improves model performance since it can help to learn compact cluster representations for fine-grained categories (Sec. 3.2.4). (3) Adding coarse-grained supervision with cross entropy loss can also boost model performance since it can contribute to representation learning. (4) Adding different filtering principles (Label, Reciprocal and Rank) can also improve model performance since they are responsible to retrieve reliable neighbors for better representation learning.

### 5.2 Accuracy of Selected Neighbors

To investigate the effect of three filtering principles on the accuracy of the retrieved neighbors, we report the accuracy (i.e., the query and neigh-

| Model | ACC | ARI | NMI |
|---|---|---|---|
| NNCL | 32.98 | 30.02 | 73.24 |
| + Multi. Neighbors | 64.89 | 52.88 | 81.14 |
| + Coarse Labels | 68.19 | 55.95 | 81.90 |
| + Label | 68.67 | 57.08 | 81.99 |
| + Reciprocal | 69.42 | 58.16 | 82.99 |
| + Rank (Ours) | 70.81 | 59.66 | 83.31 |

Table 3: Results (%) of different model variants on the HWU64 dataset. '+' means that we add the component to the previous model.

bors have the same fine-grained labels) of the retrieved neighbors in Table 4. Specifically, (1) retrieving neighbors without any filtering (**k-NN**) can yield only 50% retrieval accuracy, which can significantly affect subsequent learning since those false neighbors provide wrong supervision signal for representation learning. (2) Adding the coarse-grained label constraint (**Label**) can improve the retrieval accuracy slightly since k-NN has also learned the coarse-grained knowledge through pretraining. (3) Adding the reciprocal constraint (**Reciprocal**) can make huge improvement in retrieval accuracy since this constraint can capture bidirectional relation-

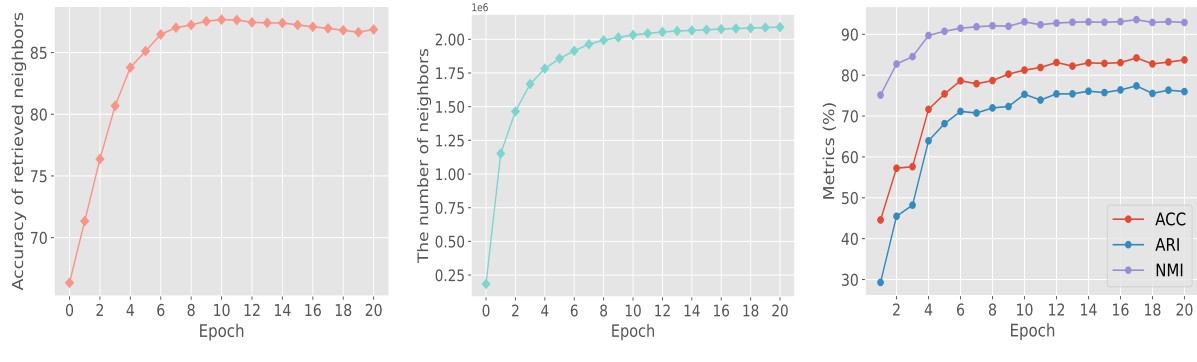

(a) Accuracy of selected neighbors.     (b) The number of selected neighbors.     (c) Model performance during training.

Figure 3: Analysis of the quality of representation learning and neighborhood retrieval.

| Model | WOS | HWU64 | CLINC |
|---|---|---|---|
| $k$-NN | 49.39 | 49.65 | 48.08 |
| +Label | 50.43 | 50.60 | 48.24 |
| +Reciprocal | 67.57 | 65.00 | 66.78 |
| +Rank | 77.34 | 66.44 | 67.28 |
| Improvement | +27.95 | +16.79 | +19.20 |

Table 4: Accuracy (%) of the retrieved neighbors.

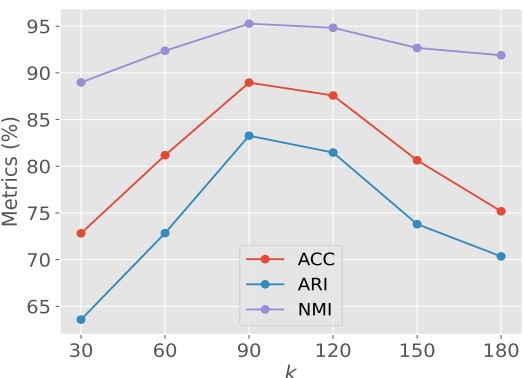

Figure 4: Influence of the number of neighbors $k$.

ships between queries and the retrieved neighbors. (4) Adding the rank statistic constraint (**Rank**) can also improve the retrieval accuracy. After pretraining, samples with the same coarse-grained labels but different fine-grained ones can be aggregated together, leading to retrieving false neighbors. The rank statistic constraint can alleviate this problem by ignoring noisy components in feature embeddings and only considering important ones that contain fine-grained information, where the important components are selected by ranking statistics.

### 5.3 Quantity and Quality Trade-off

While the filtering principles can increase the quality of neighbors by filtering out noisy ones, they can also reduce the number of retrieved neighbors. To investigate the trade-off between quantity and quality, we plot the curve of accuracy of selected neighbors (Fig. 3(a)) and the number of selected neighbors (Fig. 3(b)) on the CLINC dataset. From the figure we can see that our model can retrieve more and more neighbors with increasing accuracy during training, which means that our model can reach a trade-off between quantity and quality. We attribute this advantage to our learning paradigm where representation learning and neighborhood re-

trieval are performed alternately to bootstrap both of their performance.

### 5.4 EM Validation

To validate the intuitive EM explanation about our framework (Sec. 3.2.4), we further visualize the curve of model performance during training on the CLINC dataset in Fig. 3(c). From the figure we can see that our model gets better and better performance during training, which indicates that the quality of representation learning is gradually improved. Combined with the improved quality and quantity of the selected neighbors in Fig. 3(a) and 3(b), we can draw the conclusion that our framework can bootstrap model performance on representation learning and neighborhood retrieval iteratively through the generalized EM perspective.

### 5.5 Influence of Number of Neighbors

We investigate the influence of the number of neighbors $k$ on model performance on the CLINC dataset in Fig. 4. From the figure we can see that too large or too small $k$ can lead to poor model performance. we can also see that our model gets the best per-

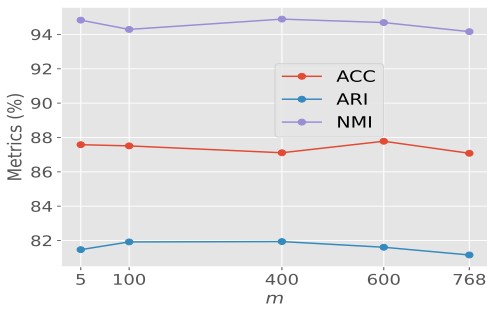

Figure 5: Influence of the number of rank dimensions.

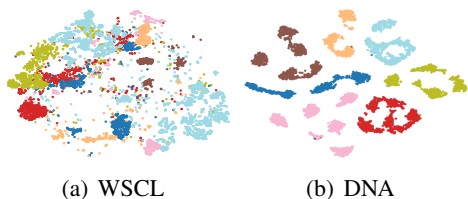

(a) WSCL          (b) DNA

Figure 6: The t-SNE visualization of embeddings.

formance when $k$ is approximately equal to the number of samples in each fine-grained category (e.g., 120 for the CLINC dataset). This is in line with our analysis in Sec. 3.2.4, since samples will gather into the center of fine-grained clusters in the ideal situation.

### 5.6 Influence of Number of Rank Dimensions

We investigate the influence of the number of rank dimensions $m$ on the CLINC dataset in Fig. 5. From the figure we can see that our model is insensitive to changes in $m$, since we only use the rank statistic constraint to select reliable neighbors for the first epoch of training. This is because rank statistic constraint is a strong constraint, which is most effective when the model performance is poor. And during training, our model can select more accurate neighbors, so we want to retrieve more neighbors to improve model's generalization ability. Furthermore, since sorting is a time-consuming process, performing rank statistic constraint only at the beginning of training can save much time and balance effectiveness and efficiency.

### 5.7 Visualization

We visualize the learned embeddings of WSCL (An et al., 2022a) and our model through t-SNE on the HWU64 dataset in Fig. 6. From the figure we can see that our model can separate different coarse-grained categories effectively. In the meanwhile, our model can maintain separability within each coarse-grained categories to facilitate the subsequent fine-grained category discovery. In summary, our model can better control both inter-class and intra-class distance between samples to facilitate the FCDC task than previous instance-level contrastive learning methods.

## 6 Conclusion

In this paper, we propose *Denoised Neighborhood Aggregation* (DNA), a self-supervised learning framework that iteratively performs representation learning and neighborhood retrieval for fine-grained category discovery. We further propose three principles to filter out false-positive neighbors for better representation learning. Then we interpret our model from a generalized EM perspective and theoretically justify that the learning objective of our model is equivalent to a clustering loss, which can encode semantic structures of data to form compact clusters. Extensive experiments on three benchmark datasets show that our model can retrieve more accurate neighbors and outperform state-of-the-art models by a large margin.

## Limitations

Even though the proposed *Denoised Neighborhood Aggregation* (DNA) framework achieves superior performance on the FCDC task, it still faces the following limitations. Firstly, DNA requires additional memory to store a queue for neighborhood retrieval and representation learning. Secondly, even though the three filtering principles can help to select more accurate neighbors, they require additional time to post-process the retrieved neighbors than traditional $k$-NN methods.

## Acknowledgments

This work was supported by National Key Research and Development Program of China (2022ZD0117102), National Natural Science Foundation of China (62293551, 62177038, 62277042, 62137002, 61721002, 61937001, 62377038). Innovation Research Team of Ministry of Education (IRT_17R86), Project of China Knowledge Centre for Engineering Science and Technology, "LENOVO-XJTU" Intelligent Industry Joint Laboratory Project.

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
