# OpenReview forum: "DNA: Denoised Neighborhood Aggregation for Fine-grained Category Discovery"
_EMNLP/2023/Conference — EMNLP 2023 Main_

### Official Review · Reviewer_Sqwo · 2023-08-05

**Typos Grammar Style And Presentation Improvements:** The paper is well written and easy to…
**Soundness:** 4

**Excitement:**

4: Strong: This paper deepens the understanding of some phenomenon or lowers the barriers to an existing research direction.

**Missing References:**

Scheirer et. al. (2016). Statistical methods for open set recognition

Badirli et. al. (2023) Classifying the unknown: Insect identification with deep hierarchical Bayesian learning

**Paper Topic And Main Contributions:**

The paper proposes a new self-supervised learning method called Denoised Neighborhood Aggregation (DNA) for fine-grained category discovery from coarsely labeled data. The main contributions are:
•	Modeling semantic structures of data instead of instance-level learning to learn compact cluster representations for fine-grained category discovery.
•	Introducing 3 principles (label, reciprocal, rank constraints) to filter false neighbors for a more accurate representation for the  query.
•	Providing a theoretical analysis to show the learning objective mimics the clustering loss to form compact clusters.
•	Demonstrating significant performance boost on 3 benchmark datasets, with improved neighbor accuracy over state of the art methods.


**Questions For The Authors:**

A: What is the impact of the queue size and momentum update coefficient on model performance?
B: Can authors try to validate the argument in L62-64 with some theoretical or experimental  evidence? "Despite the improved performance, these instance-based methods fail to encode cluster-level semantic structures of data"
C: Can you provide more ablation results to demonstrate the individual impact of the 3 filtering principles?
D: Could you please elaborate why "uniform" part in objective function is important?

**Reasons To Accept:**

Authors tackle an important and practical problem of discovering fine-grained categories from coarsely labeled data.
•	The three principles for filtering false neighbors are reasonable and effective as evidenced by improved neighbor accuracy and reflected performance boost on clustering metrics.
•	Their simple theoretical analysis provides good justification and intuition behind the objective function for alignment part. Interpretation as generalized EM is insightful.
•	Strong experimental results across three datasets demonstrate effectiveness of the model, significantly outperforming prior state-of-the-art.
•	The problem being addressed is relevant, with applications in areas like computer vision and NLP.


**Reasons To Reject:**

•	The proposed ideas like loss function, dynamic queue and nearest neighbor help, are updated yet incremental improvements over prior work like nearest neighbor contrastive learning. That being said, it is good to see successful implementations from cross-domains (CV to NLP)
•	Additional memory is required to store the queue for neighborhood retrieval. Filtering principles increase training time.
•	More details on hyper-parameter choice/tuning would be helpful to better analyze the sensitivity of the model to different parameter choices, like k , rank dimensions etc.

**Reproducibility:**

4: Could mostly reproduce the results, but there may be some variation because of sample variance or minor variations in their interpretation of the protocol or method.

**Reviewer Confidence:**

4: Quite sure. I tried to check the important points carefully. It's unlikely, though conceivable, that I missed something that should affect my ratings.

---

> ### Author Rebuttal · Authors · 2023-08-28
>
> Thank you for the constructive comments and suggestions. The responses to your main concerns are listed below.
>
> [Q1] The proposed ideas like loss function, dynamic queue and nearest neighbor help, are updated yet incremental improvements over prior work like nearest neighbor contrastive learning. That being said, it is good to see successful implementations from cross-domains (CV to NLP).
>
> [R1] Our model is based on the nearest neighbor contrastive learning, and our novelty lies in following three aspects.
> + We analyze the problems of existing FCDC methods and extend the traditional nearest neighbor contrastive learning to multiple neighbors to solve the problem, and the multiple-neighbor strategy is effective for FCDC (please refer to Table 3 in the paper).
> + We theoretically prove that the learning objective of our multiple-neighbor contrastive learning is equivalent to a clustering loss, which provides a theoretical foundation for why our model can learn compact cluster representations.
> + We propose three principles to filter out false neighbors for better representation learning, where the quality of neighbors is not addressed by pervious methods.
>
> [Q2] Additional memory is required to store the queue for neighborhood retrieval. Filtering principles increase training time.
>
> [R2] Our model can minimize the impact of memory usage by maintaining a smaller queue without losing much model performance (please refer to [R4]). As for training time, we report the compared training time in Table 6 in our paper, and the additional training time is minimal. And our model can further reduce the training time by updating neighborhood relations every five epochs and get comparable training efficiency as previous methods (85.2 s/epoch vs. 67.0 s/epoch).
>
> [Q3] More details on hyper-parameter choice/tuning would be helpful to better analyze the sensitivity of the model to different parameter choices, like k , rank dimensions etc.
>
> [R3] We included experiments and analysis of some hyperparameters in our original paper (e.g., k in Section 5.5 and Fig. 4, rank dimensions in Appendix A.5 and Fig. 7), as well as the impact of queue size and momentum coefficient in [R4] below.
>
>
> [Q4] What is the impact of the queue size and momentum update coefficient on model performance?
>
> [R4] For the impact of queue size, we conducted experiments with different ratio of data on the HWU64 dataset, and the results are shown below.
>
> | Ratio | ACC | ARI | NMI |
> | ---- | --- | --- | --- |
> | 30% | 65.16 | 55.38 | 81.98 |
> | 50% | 68.52 | 57.93 | 82.31 |
> | 70% | 70.62 | 60.00 | 83.01 |
> | 90% | 69.55 | 58.36 | 82.61 |
> | 100% | 70.81 | 59.66 | 83.31 |
>
> From the results we can see that our model can get comparable performance even with a smaller queue.
>
> We also conducted experiments for momentum coefficient on the HWU64 dataset, and the results are shown below.
>
> | Coefficient | ACC | ARI | NMI |
> | ---- | --- | --- | --- |
> | 0 (w/o momentum) | 68.87 | 57.86 | 82.46 |
> | 0.3 | 70.16 | 59.96 | 83.21 |
> | 0.5 | 69.98 | 59.84| 83.28 |
> | 0.7 | 69.56 | 59.99 | 84.40 |
> | 0.9 | 68.95 | 58.75 | 82.55 |
> | 0.99 | 70.81 | 59.66 | 83.31 |
>
> From the results we can see that our model is insensitive to the momentum coefficient, which can show the robustness of our model.
>
>
> [Q5] Can authors try to validate the argument in L62-64 with some theoretical or experimental evidence? "Despite the improved performance, these instance-based methods fail to encode cluster-level semantic structures of data"
>
> [R5] We can validate the argument with both theoretical and experimental evidence.
> + Theoretically, [1] proves that compared to instance features, class centers (prototypes) have  more mutual information with the class labels, which means that pulling samples closer to class centers (prototypes) can help to capture cluster-level semantics between data with the same fine-grained category and learn better cluster representations. Furthermore, our theoretical analysis in Sec. 3.2.4 also proves that our model can capture semantic structures between data by pulling samples with high semantic similarity closer and clustering these samples to form compact clusters.
> + Experimentally, we have conducted experiments to validate the argument. Since good feature representations require samples of the same category to have high semantic similarity and samples of different categories to have low semantic similarity, we define a score as average semantic similarity between samples from the same fine-grained category **minus** average semantic similarity between samples from different fine-grained categories. A higher score indicates that the model can better capture semantic similarities between data. The results on three datasets are shown below.
>
> | Dataset | WSCL | Ours |
> | ---- | --- | --- |
> | WOS   | 0.4090 | 0.5564 |
> | CLINC  | 0.5676 | 0.7197 |
> | HWU64 | 0.5732 | 0.5897 |
>
> From the results we can see that our model gets higher scores on all datasets, which means that our model can better capture semantic relationships between data compared to previous SOTA methods.
>
>
> [Q6] Can you provide more ablation results to demonstrate the individual impact of the 3 filtering principles?
>
> [R6] For ablation study in Table 3, we conducted individual impact experiments. We define the multi-neighbor contrastive learning with coarse supervision as the base model.
>
> | Model | ACC | ARI | NMI |
> | ---- | --- | --- | --- |
> | Base             | 68.19 | 55.95 | 81.90 |
> | Base + Label      | 68.67 | 57.08 | 81.99 |
> | Base + Reciprocal  | 68.81 | 57.64 | 82.35 |
> | Base + Rank       |70.11 | 60.10 | 83.22 |
>
> From the results we can see that individual filtering principle can also improve our model performance, which can show the individual effectiveness of our filtering principles.
>
> For accuracy of retrieved neighbors in Table 4, we also conducted individual impact experiments.
>
> | Model | WOS | HWU64 | CLINC |
> | ---- | --- | --- | --- |
> | k-NN             | 49.39 | 49.65 | 48.08 |
> | k-NN + Label      | 50.43 | 50.60 | 48.24 |
> | k-NN + Reciprocal  | 66.69 | 65.41 | 66.32 |
> | k-NN + Rank        |72.10 | 60.12 | 55.22 |
>
> From the results we can see that individual filtering principle can also improve the accuracy of retrieved neighbors, which means that our filtering principles are effective. Overall, our filtering principles are effective, even when using the individual filtering principle.
>
>
> [Q7] Could you please elaborate why "uniform" part in objective function is important?
>
> [R7] The "uniform" part corresponds to negative keys in contrastive learning. As theoretically and experimentally demonstrated in [2], the "uniform" part is responsible for preserving maximal information of data and making features uniformly distributing on the feature hypersphere, which is vital for learning good representations. Without this term, all sample features will collapse to a single point in the feature hypersphere to maximize the alignment part, which can lead the model to fail to learn any useful representations.
>
> [Q8] Missing References
>
> [R8] Thanks for your suggestions, we will add these references in the revised version.
>
>
>
> References
>
> [1] Li J, Zhou P, Xiong C, et al. Prototypical Contrastive Learning of Unsupervised Representations[C]//International Conference on Learning Representations. 2020.
>
> [2] Wang T, Isola P. Understanding contrastive representation learning through alignment and uniformity on the hypersphere[C]//International Conference on Machine Learning. PMLR, 2020: 9929-9939.

---

### Official Review · Reviewer_Mz1B · 2023-08-05

**Soundness:** 4

**Excitement:**

4: Strong: This paper deepens the understanding of some phenomenon or lowers the barriers to an existing research direction.

**Paper Topic And Main Contributions:**

This paper proposes a novel approach to address the problem of learning compact cluster representations for the Fully Connected Dense Clustering task. The paper addresses the problem of learning compact cluster representations for the FCDC task. It proposes a novel framework, Denoised Neighborhood Aggregation, and provides a theoretical foundation for its effectiveness. The experimental evaluations demonstrate that the proposed method outperforms existing approaches in both the FCDC task and neighbor retrieval accuracy, which further validates the theoretical claims and establishes the paper's contributions to the field.

**Reasons To Accept:**

1) The paper introduces a fresh perspective by advocating the modeling of semantic structures in data for learning more compact cluster representations.
2) The proposed Denoised Neighborhood Aggregation framework presents a novel self-supervised approach for capturing semantic similarities between data points and aggregating information from neighboring data points.
3) The paper provides a solid theoretical foundation by interpreting the proposed framework from a generalized Expectation-Maximization perspective. The theoretical proof, which establishes the equivalence of the learning objective to a clustering loss, adds credibility to the proposed approach and its potential impact on NLP research.
4) The FCDC task and representation learning are crucial components of many NLP applications. By improving the efficiency and accuracy of clustering methods, the paper's contributions can have broad applications, such as document clustering, topic modeling, sentiment analysis, and information retrieval, which are of utmost importance in NLP research and applications.
5) The proposed principles for filtering out false neighbors and the theoretical interpretation from an EM perspective can serve as a basis for further research and inspire the development of new clustering and representation learning techniques in NLP.

**Reasons To Reject:**

1) The paper's experiments should aim to conduct a broader range of ablation studies on multiple datasets. Currently, the experiments seem to be limited to only one or a few datasets, which may lead to overfitting and limit the generalizability of the proposed approach to other real-world scenarios.
2) The paper's visual experiments should incorporate control groups. Presenting results without control groups may lead to difficulties in interpreting the effectiveness of the proposed method compared to existing or alternative approaches. The inclusion of control groups helps establish a solid basis for making valid conclusions.

**Reproducibility:**

4: Could mostly reproduce the results, but there may be some variation because of sample variance or minor variations in their interpretation of the protocol or method.

**Reviewer Confidence:**

3: Pretty sure, but there's a chance I missed something. Although I have a good feel for this area in general, I did not carefully check the paper's details, e.g., the math, experimental design, or novelty.

---

> ### Author Rebuttal · Authors · 2023-08-28
>
> Thanks for your constructive comments and suggestions. The responses to your concerns are listed below.
>
> [Q1] The paper's experiments should aim to conduct a broader range of ablation studies on multiple datasets.
>
> [R1] Thanks for your suggestions, we only report results on one dataset due to the page limit. We have added ablation studies on other two datasets. The results are listed below.
>
> **CLINC Dataset**
> | Model | ACC | ARI |NMI |
> | --- | --- |  --- |  --- |
> | NNCL |                   17.42 | 13.93 | 67.56 |
> | \+ Multi. Neighbors |       85.31 | 78.33 | 92.76 |
> | \+ Coarse Labels  |        85.69 | 80.23 | 93.08 |
> | \+ Label  |               86.89 | 81.53 | 94.13 |
> | \+ Reciprocal  |           87.46 | 81.29 | 94.09 |
> | \+ Rank (Ours)  |          87.66 | 81.82 | 94.69 |
>
> **WOS Dataset**
> | Model | ACC | ARI |NMI |
> | --- | --- | --- | --- |
> | NNCL |                    29.64 | 28.51 | 61.37 |
> | \+ Multi. Neighbors |        71.02 | 61.32 | 75.18 |
> | \+ Coarse Labels |           72.23 | 62.02 | 75.89 |
> | \+ Label |                  72.85 | 61.50 | 75.89 |
> | \+ Reciprocal |              73.93 | 62.41 | 76.61 |
> | \+ Rank (Ours) |             74.57 | 63.30 | 76.86 |
>
> For other experiments (e.g., Fig 3, 4, 5), we have also performed experiments on other two datasets, and the results demonstrate the similar trend as the results reported in the paper, which can validate the conclusions drawn in our paper and show effectiveness and generalizability of our model. However, we cannot show these images here because of the textual constraints of openreview platform. We will add these details and conclusions in the revised version.
>
> [Q2] The paper's visual experiments should incorporate control groups.
>
> [R2] Thanks for your suggestions, we only report our visual results due to the page limit. We have added visual experiments compared to existing SOTA approaches (WSCL and SimCSE + CE). Compared to these methods, our model can learn more separable fine-grained clusters, which is consistent with the clustering metrics in Table 2. Limited by the openreview platform, we cannot show these images during rebuttal, but we will include these control groups in the revised version.

---

### Official Review · Reviewer_1nk5 · 2023-08-07

**Soundness:** 4

**Excitement:**

4: Strong: This paper deepens the understanding of some phenomenon or lowers the barriers to an existing research direction.

**Paper Topic And Main Contributions:**

- This paper focuses on the task of FCDC.
- This paper proposes a novel framework for FCDC, called DNA.
- The proposed framework is interpreted from a generalized EM perspective, and is proved that the learning object is equivalent to a clustering loss.
- The experimental results show that the proposed framework outperforms other methods on three benchmark datasets.

**Reasons To Accept:**

- This paper is well-written and is easy to follow.
- Compared with SOTA methods, the experimental results of the proposed framework show large improvements, which demonstrates the effectiveness of DNA.
- The proposed three principles to filter out false-positive neighbors are insightful.

**Reasons To Reject:**

- In lines 071-074, what is the definition of "compact" representation? Why more compact cluster representations mean more separable fine-grained categories?
- How to solve the situation that no neighbors left after filtering by three principles?
- The features in query and the features in queue are extracted from different encoder, why the similarity between them can be directly computed?
- In Table 2, Ancor -> Anchor

**Reproducibility:**

4: Could mostly reproduce the results, but there may be some variation because of sample variance or minor variations in their interpretation of the protocol or method.

**Reviewer Confidence:**

3: Pretty sure, but there's a chance I missed something. Although I have a good feel for this area in general, I did not carefully check the paper's details, e.g., the math, experimental design, or novelty.

---

> ### Author Rebuttal · Authors · 2023-08-28
>
> Thank you for the constructive comments and suggestions. The responses to your concerns are listed below.
>
> [Q1] What is the definition of "compact" representation? Why more compact cluster representations mean more separable fine-grained categories?
>
> [R1] As analyzed in Section 3.2.4, we define ‘compact’ as samples with the same category are compactly clustered into the center of category and away from samples from different categories, which means smaller intra-class distance and larger inter-class distance. Since samples distributed around decision boundaries are easily misclassified into other categories, distributing samples near the category center compactly can avoid overlapping decision boundaries of different categories and make these categories more separable. Similar observations have also been made by other researchers such as the phenomenon of Neural Collapse [1].
>
>
> [Q2] How to solve the situation that no neighbors left after filtering by three principles?
>
> [R2] We solve this problem by ‘back off’ strategy, which means that we can  implement only parts of the filtering principles rather than all of them  to make sure at least some neighbors are left. Empirically we found that the case of no neighbors is unlikely to happen, especially after the first epoch as the number of neighbors is usually large (Fig. 3(b) in the paper). We will add this detail in the revised version.
>
> [Q3] The features in query and the features in queue are extracted from different encoder, why the similarity between them can be directly computed?
>
> [R3] Since features in the queue are updated during training, we use the slowly changing momentum encoder to update features in the queue to avoid these features to differ too much or sudden when neighborhood retrieval (before training) and representation learning (during training), especially at the last few iterations of an epoch, where the momentum encoder can make the training more stable. Furthermore, both encoders are initialized identically and the momentum encoder is updated according to the parameters of the encoder (Eq. 1 in the paper), so the difference between the two encoders is minimal. The momentum encoder techniques have been proven effective and widely used in both CV [1] and NLP [2]. Furthermore, we also perform experiments to validate the effectiveness of the momentum encoder.
> | Dataset | w/ momentum | w/o momentum |
> | ---- | --- | --- |
> | CLINC | {ACC: 87.66, ARI: 81.82, NMI: 94.69} | {ACC: 86.44, ARI: 80.22, NMI: 93.26} |
> | WOS |  {ACC: 74.57, ARI: 63.30, NMI: 76.86} | {ACC: 74.28, ARI: 61.77, NMI: 75.90} |
> | HWU64 | {ACC: 70.81, ARI: 59.66, NMI: 83.31} | {ACC: 68.87, ARI: 57.86, NMI: 82.46} |
>
> [Q4] In Table 2, Ancor -> Anchor.
>
> [R4] Thanks for pointing out this typo, we will correct it in the revised version.
>
> References
>
> [1] Kothapalli V, Rasromani E, Awatramani V. Neural collapse: A review on modelling principles and generalization[J]. arXiv preprint arXiv:2206.04041, 2022.
>
> [2] He K, Fan H, Wu Y, et al. Momentum contrast for unsupervised visual representation learning[C]//Proceedings of the IEEE/CVF conference on computer vision and pattern recognition. 2020: 9729-9738.
>
> [3] An W, Tian F, Chen P, et al. Fine-grained Category Discovery under Coarse-grained supervision with Hierarchical Weighted Self-contrastive Learning[C]//Proceedings of the 2022 Conference on Empirical Methods in Natural Language Processing. 2022: 1314-1323.

---

### Meta-Review · Area_Chair_YaNz · 2023-09-16

**Recommendation:** 5

**Metareview:**

This paper delves into the problem of fine-grained category discovery and introduces denoised neighborhood aggregation, a self-supervised framework aimed at encoding semantic structures into the embedding space. The paper provides both theoretical analysis and empirical experiments to demonstrate the effectiveness of the proposed model.

The consensus among all reviewers is that this paper exhibits strong soundness and excitement. However, there are several concerns that need further attention from the authors, including:

1. Definition Clarification: Some term definitions are found to be confusing and would benefit from clearer explanations.
2. Boundary Settings: For example, how to solve the situation without any neighbors. The authors can give more explanations.
3. Ablation Study: A broader range of ablation studies on multiple datasets should be conducted to make the experiments more convincing.
4. Hyper-parameter Settings: The paper lacks sufficient details regarding the choice of hyper-parameter settings, which should be provided for transparency.

In conclusion, this paper offers strong quality and novelty, with the potential to contribute significantly to the research community. To further improve its quality, I recommend that the authors diligently address the concerns raised by the reviewers during their final revision.

---

### Decision · Program_Chairs · 2023-10-07

**Decision:**

Accept-Main

**Comment:**

This paper delves into the problem of fine-grained category discovery and introduces denoised neighborhood aggregation, a self-supervised framework aimed at encoding semantic structures into the embedding space. The paper provides both theoretical analysis and empirical experiments to demonstrate the effectiveness of the proposed model.

The consensus among all reviewers is that this paper exhibits strong soundness and excitement. However, there are several concerns that need further attention from the authors, including:

1. Definition Clarification: Some term definitions are found to be confusing and would benefit from clearer explanations.
2. Boundary Settings: For example, how to solve the situation without any neighbors. The authors can give more explanations.
3. Ablation Study: A broader range of ablation studies on multiple datasets should be conducted to make the experiments more convincing.
4. Hyper-parameter Settings: The paper lacks sufficient details regarding the choice of hyper-parameter settings, which should be provided for transparency.

In conclusion, this paper offers strong quality and novelty, with the potential to contribute significantly to the research community. To further improve its quality, I recommend that the authors diligently address the concerns raised by the reviewers during their final revision.